# Evaluation of Genistein as a Mitochondrial Modulator and Its Effects on Sperm Quality

**DOI:** 10.3390/ijms241814260

**Published:** 2023-09-19

**Authors:** Marilia Ferigolo, Jessica Nardi, Natália Freddo, Alessandra Ferramosca, Vincenzo Zara, Eliane Dallegrave, Mateus Belmonte Macedo, Sarah Eller, Ana Paula de Oliveira, Inara Carbonera Biazus, Francieli Ubirajara India Amaral, Luciana Grazziotin Rossato-Grando

**Affiliations:** 1Graduate Program in Bioexperimentation, University of Passo Fundo, BR 285 Km 292,7, Campus I, Passo Fundo 99052-900, Brazil; 190675@upf.br (M.F.); jessicanardi@upf.br (J.N.); 135284@upf.br (N.F.); 59933@upf.br (F.U.I.A.); rossatoluciana@upf.br (L.G.R.-G.); 2Institute of Health, Faculty of Pharmacy, University of Passo Fundo, BR 285 Km 292,7, Campus I, Passo Fundo 99052-900, Brazil; 191294@upf.br (A.P.d.O.); 184172@upf.br (I.C.B.); 3Department of Biological and Environmental Sciences and Technologies, University of Salento, I-73100 Lecce, Italy; vincenzo.zara@unisalento.it; 4Department of Pharmacosciences, Federal University of Health Sciences of Porto Alegre (UFCSPA), Sarmento Leite Street, 245, Porto Alegre 90050-170, Brazil; elianedal@ufcspa.edu.br (E.D.);; 5Laboratory of Research in Toxicology, Federal University of Health Sciences of Porto Alegre (UFCSPA), Sarmento Leite Street, 245, Porto Alegre 90050-170, Brazil; mateusbmacedo@gmail.com

**Keywords:** genistein, isoflavones, phytoestrogens, spermatogenesis, mitochondria

## Abstract

Phytoestrogens, such as isoflavones, are bioactive compounds found in plants with defense and protection functions. In the human body, they simulate the behavior of the hormone estradiol and can modulate the function of the male hypothalamic–pituitary–gonadal axis. This study aims to describe the effects of genistein on sperm quality of Wistar rats (male/adult) after a short oral administration protocol (50 mg/day, for 5 days), focusing on mitochondrial function. No signs of toxicity were observed in the animals during the period. The testicular mass of rats from the genistein-treated group was lower than that from the control group. Isoflavone increased the number of viable Leydig and Sertoli cells, spermatogonia, and primary spermatocytes in the treated group. The rounded spermatid count was similar to the control group, and a decrease in elongated spermatids was observed in the treated group. Genistein treatment increased plasma testosterone levels in the treated group. To the best of our knowledge, this is the first report of an in vivo short protocol demonstrating that genistein administration stimulates the overall oxygen consumption in rat seminal samples. Therefore, genistein induced a pro-spermatogenesis effect, enhanced plasma testosterone levels, and increased oxygen consumption, improving sperm mitochondrial efficiency. Similar protocols can be explored in animal and human infertility issues.

## 1. Introduction

Phytoestrogens are natural non-steroidal phenolic compounds from plants and can be divided into two main groups: flavonoids and non-flavonoids. Flavonoids include isoflavones, coumestans, and prenylflavonoids [1]. In plants, phytoestrogens play an important role in plant–microbiota interactions, including defense and symbiosis [1]. Humans are exposed to a range of bioactive flavonoid substances, including natural phytoestrogens, through their diet [2,3,4]. Isoflavones are present in soy and other legumes, such as beans, alfalfa, and chickpeas, and in smaller amounts in fruits, vegetables, and nuts [1].

Isoflavones, including genistein, are found in soy and soy-based foods. The isoflavone content in soybeans varies slightly according to the crop and growing conditions but usually represents 0.2–0.3% of dry weight [5]. Soybeans contain particularly high amounts of isoflavones, which makes them advantageous and easy to consume [2]. Although the suggested daily intake of isoflavones has not yet been established, the US Food and Drug Administration (FDA) recommends that an intake of 25 g of soy per day is considered safe [6].

Genistein exerts several biological activities. As a phytoestrogen, it acts as an estrogen agonist or antagonist in mammals [7]. Its biological activity is triggered by binding to estradiol receptors (ER1 and ER2) [4]. The affinity of genistein is about 20–30 times higher for ER2 [8]. Although isoflavones have low affinity for estradiol hormone receptors, they have proven to be remarkably efficient in stimulating sperm function [9]. These receptors are present with specific selectivity in several estradiol-responsive tissues, including those that are part of the hypothalamic–pituitary–gonadal axis, responsible for the reproductive function [4]. Additionally, the effects of genistein appear to be dose-dependent and vary by individual [10,11].

The health properties of genistein appear to go beyond its estrogen-like activity. Examples are the reported preclinical pharmacological and metabolic activities of genistein as an important source of antioxidant, anti-inflammatory, and antimicrobial properties [7].

Information on the effect of bioactive plant molecules on sperm quality is extensive but, at the same time, fragmented or inconsistent. This is mainly due to the wide variety of experimental conditions used in the studies carried out on the subject in gametes of different species [10]. When semen is incubated in vitro with low concentrations of genistein, an improvement in the quality of sperm is observed [10]. In addition, higher amounts of isoflavone can be toxic to sperm, causing a cellular imbalance and loss of function [3,9,10]. In vivo, our research group showed that, after chronic exposure throughout the pre-pubertal period to an isoflavone-rich diet, there was a decrease in testosterone and sperm quality when rats reached adolescence [12].

Mitochondria are key organelles in crucial sperm functions, such as motility, hyperactivation, capacitation, acrosome reaction, and fertilization [13,14,15,16,17]. Any structural or functional dysfunction of sperm mitochondria results in increased production of reactive oxygen species and decreased energy production, leading to sperm DNA damage and impaired sperm motility and semen parameters, consequently reducing male fertility [18]. Therefore, sperm mitochondrial functionality assessment is frequently used as an important screening tool for evaluating not only sperm quality but also the effects of various compounds, especially those with a hormone-like activity such as isoflavone [10,19,20,21,22,23,24].

The purpose of this study is to understand the effects of the isoflavone genistein on the quality of spermatozoa of Wistar rats, focusing on mitochondrial function.

## 2. Results

### 2.1. Water and Diet Consumptions

We did not observe any clinical signs suggestive of systemic toxicity during the study. Moreover, there was no significant difference in relative body mass gain between groups as well as in food consumption (Figure 1A).

However, there was a higher relative water consumption by the group treated with genistein from day 4 (*p* < 0.01) (Figure 1B).

### 2.2. Relative Mass of the Sexual Organs

The relative mass of testicles from the genistein group was significantly lower than the control group (Figure 2) (*p* < 0.05). We did not observe significant difference in the relative mass of other sexual organs (epididymis, epididymal tail, full seminal vesicle, empty seminal vesicle, or ventral prostate).

### 2.3. Plasma Testosterone Levels

Rats treated with genistein presented significantly higher plasma testosterone levels compared to control animals (*p* < 0.05) (Figure 3).

### 2.4. Evaluation of Seminiferous Tubules

In the microscopy analysis of seminiferous tubules, a seminiferous tubule cross-section was conducted, and a scan was performed in 10 slides (*n* = 5 animals each group). The results were presented as the mean of the measurements of 100 seminiferous tubules from each rat. The comparison of the diameter of the seminiferous tubules between groups had significant differences (*p* < 0.05) (Figure 4A). The thickness of the seminiferous tubules did not present statistical differences between groups (Figure 4B).

### 2.5. Morphological Cellular Evaluation

Genistein increased the number of viable Leydig and Sertoli cells, spermatogonia, and primary spermatocyte cells. The number of rounded-shaped spermatids was similar to control levels and a decrease in elongated spermatids was observed (Table 1 and Figure 5). Cellular changes were scanned. The results are presented in Table 1 as mean ± standard deviation from five animals for each group (five seminiferous tubules/rat).

### 2.6. Mitochondrial Functionality Assays

#### Epididymal Sperm Mitochondria Respiration Efficiency

To evaluate mitochondrial functionality, we isolated sperm cells from rat cauda epididymis. Genistein significantly improved mitochondrial respiration efficiency, as suggested by RCR (respiratory control ratio) values, which is an index of mitochondrial functionality (Table 2). The data reported in Table 2 show that genistein causes a slight yet significant increase in the V_3_ values (rate of oxygen uptake measured in the presence of respiratory substrates + ADP, i.e., state three of respiration) and a decrease of about 20% in the V_4_ values (rate of oxygen uptake measured with substrates alone, i.e., state four of respiration). Thus, the RCR values increase by about 30% in the genistein group.

When the activity of the single components of the respiratory chain (complexes I–IV) was assayed spectrophotometrically, we observed that genistein caused only a small enhance (about 10%) in the activity of mitochondrial complex III (Table 3).

To understand the biological implications of the effects of genistein on the mitochondrial respiratory chain, we measured adenosine triphosphate (ATP) concentration and oxidative damage in sperm samples from the two groups of animals. Genistein caused a significant increase (about 15%) in ATP content and a significant decrease (12%) in the lipid hydroperoxide (LPO) levels (Figure 6).

## 3. Discussion

We demonstrate that acute exposure to genistein at the relevant dose of 50 mg/day over a short period of 5 days can improve sperm mitochondrial respiration efficiency. Genistein induced an early pro-spermatogenesis effect evidenced by increases in the viable Leydig and Sertoli cells, spermatogonia, and primary spermatocyte cells. The increased Leydig cells are related to increased testosterone levels, also observed in the present study. However, these changes do not result in increased sperm count as an increase in degenerated primary spermatocytes was also observed compared to control animals. These changes resulted in a number of rounded-shaped spermatids similar to control and decreased number of elongated spermatids. Even so, sperm showed a better mitochondrial profile.

The function of mitochondria as energy suppliers is certainly fundamental for sperm motility. Indeed, defects in the mitochondrial ultrastructure of spermatozoa are associated with asthenozoospermia [16]. Also, mitochondrial oxygen consumption is considered a central parameter of mitochondrial function and is positively correlated with sperm motility and vitality [15]. Thus, recent research in sperm physiology is focusing on this important organelle as a biomarker of sperm health and fertility [15,18].

To the best of our knowledge, this is the first report of an in vivo short protocol demonstrating that genistein administration stimulates the overall oxygen consumption in rat seminal samples. A short protocol with relevant effects is desirable because it is related to decreased costs, increased treatment adherence, and, consequently, success. This result is in line with what was recently shown in vitro where genistein had a positive effect at the lowest concentration (0.1 nmol/L) after incubation with semen from asthenozoospermic individuals, with a motility of about 25%. Along with genistein, luteolin and quercetin improved mitochondrial function in asthenozoospermic samples [10]. Genistein alone was able to improve mitochondrial function by increasing the efficiency of mitochondrial respiration (34%) [10].

Additionally, our study showed that genistein causes a small increase (about 10%) in the activity of mitochondrial complex III, which plays a central role in the ROS (reactive oxygen species) production in sperm cells [25]. However, our data showed a parallel increase in the cellular levels of ATP, a molecule that provides energy to cells for various functions, and a decrease in oxidative damage. Interestingly, an interaction between genistein and mitochondrial complex III has already been proposed [26].

Sperm mitochondrial dysfunction is implicated in the pathogenesis of seminal oxidative stress, a key element responsible for many cases of “apparently” idiopathic male infertility. Since mitochondria play a major role in metabolism, energy production, and sperm motility, they are the main generators of ROS [25]. An imbalance between ROS production and antioxidant mechanisms can be extremely harmful to sperm, especially mature ones. These cells are sensitive to oxidation because they lack adequate repair mechanisms and possess inadequate antioxidant capacity. In contrast, at physiological concentrations, ROS trigger several important reproductive controls, such as sperm capacitation, hyperactivation, acrosome reaction, and oocyte fusion [18]. More recent studies are changing the dogma that ROS is a negative indicator of sperm function, indicating that ROS production may also reflect intense mitochondrial activity, leading to increased sperm function [15,18].

The seminiferous epithelial cycle is a complete progressive series of changes in cell associations (stages) in a segment (portion of a seminiferous tubule occupied by a cell association). About 4 or 4.5 cycles are necessary for a complete spermatogenesis. In Wistar rats, it takes around 13 days. This is important to understand normal spermatogenesis or what might go wrong with it [27]. Accordingly, the present study demonstrated that administration of genistein during a five-day period promoted changes in cells’ development inside the seminiferous tubules.

Studies show that the number of Sertoli cells per testis is the main factor in determining sperm production. Thus, the number of germ cells supported by a single Sertoli cell is the best indicator of their functional efficiency [28]. Our study presented the increased number of Sertoli cells as the best indication of efficiency and support capacity. It enhanced the number of germ cells, demonstrating a positive effect of genistein treatment.

Rats treated with genistein presented significant changes in seminal cell count. The number of viable Leydig cells increased in greater proportion than the degenerated ones (which did not present a statistically significant difference). We hypothesize that an increased number of Leydig cells is related with the increase in testosterone production, which in turn stimulates the production of Sertoli cells, consequently increasing its proportion. The increase in Sertoli cells was observed in both viable and degenerated cells, and the reason remains unclear. A complex series of molecular events and proper interactions between germ cells, Sertoli cells (which coordinate and provide structural support to developing germ cells), epithelial tube cells, and the integrity of the blood–testis barrier are required for successful spermatogenesis [29]. Additionally, mitochondrial energy synthesis also contributes to steroid hormone biosynthesis in Leydig cells [13].

Spermatogenesis occurs in the germinal epithelium and is controlled by hormones by the pituitary gland, through hypothalamic–pituitary–gonadal axis action. Leydig and Sertoli cells are responsible for producing androgens and estrogens, which regulate the spermatogenesis process [4]. Leydig cells contain surface receptors for luteinizing hormone (LH) and respond to stimulation by producing and releasing testosterone. Then, testosterone diffuses into the seminiferous tubule, where receptors for testosterone are present in Sertoli cells [27]. So, rats treated with genistein in our study had a significant increase in plasmatic testosterone levels compared to the control group. The increase in this hormone may have happened due to the fact of the connection that occurs between testosterone and Sertoli cells inside the testicles since there was an increase in the number of those cells.

Therefore, an imbalance in androgen or estrogen production can affect spermatogenesis [4,11]. Recent findings defend flavonoids and isoflavones such as chrysin, apigenin, luteolin, quercetin, and daidzein along with genistein, which may be effective in delaying the initiation of late-onset hypogonadism associated with aging in males to improve testosterone production, contributing to normal spermatogenesis, male fertility, and preventing age-related degenerative diseases associated with testosterone deficiency [30].

Testicular weight is an important parameter in the andrological evaluation of mammals since it reflects normality and allows inferring the sperm production rate [28]. In this study, the relative mass of testicles from the genistein group was significantly lower than the control group after 5 days of exposure to genistein. This change was accompanied by a decrease in the total size of the seminiferous tubules of the genistein-treated group. Decreases in the testis weight of adult male rats exposed to mixtures of isoflavones were previously reported. Rats receiving genistein + daidzein (17 mg/kg/day and 12 mg/kg/day, respectively) or higher doses (170 mg mg/kg/day + 120 mg/kg/day, respectively) presented decreased testis relative weight after 12 weeks [11]. In our study, testicle changes were observed after such a short exposure time. Changes in sex steroids hormones and sperm quality can occur in the testicles due to variations in the size and morphological cellular alterations [11].

Regarding the seminiferous tubule diameter, morphometric analysis of the diameter of the seminiferous tubules and the thickness of the germinal epithelium are positively correlated with diameter and spermatogenic activity [31]. Therefore, a study has demonstrated that the consumption of isoflavones causes a reduction in the seminiferous tubules diameter and germinal epithelium height that could affect sperm quality parameters [32].

Moreover, there was no significant difference in the body mass gain between groups as well as in food consumption. Also, no clinical signs suggestive of systemic toxicity during the study were observed between groups. Furthermore, in our work, there was higher water consumption by the genistein-treated group. This change is probably related to the bitter taste of the genistein solution. However, no differences in food and water consumption were observed in one recent study with *Wistar* rats among experimental groups except during the first weeks of experimentation. The consumption of isoflavones at low or high doses did not affect body weight significantly in groups treated with isoflavones (genistein + daidzein) compared with a control group. Further, after the 12th week of experimentation, the group that was administered low doses (17 mg/kg/day and 12 mg/kg/day, respectively) demonstrated a significant loss in body weight compared to the control group, while those receiving high doses (170 mg mg/kg/day + 120 mg/kg/day, respectively) showed a significant gain in body weight [11].

Genistein is the isoflavone found in the greatest amount in soybean and is widely consumed, either because it is a traditional food or because of its interesting biological effects in the body. Furthermore, the decision to use it alone for the study is also due to the fact that, in the literature, there are a range of studies using conjugated isoflavones. In the present study, we present interesting and expressive findings when associating it with male fertility. For perspective, the results presented here suggest that short protocols of genistein supplementation can be tested in order to improve the success of human and/or animal fertilization.

## 4. Materials and Methods

### 4.1. Commercial Products

Genistein (purity ≥ 98%) acquired by LEMMA Supply© (BR21052501; Sao Paulo, Brazil), was used in the experiment.

Testosterone-*d*_3_ (internal standard; IS) at a concentration of 100 µg/mL was obtained from Cerilliant Analytical Reference Standards (Round Rock, TX, USA) (T-046). HPLC-grade acetonitrile and formic acid were purchased from Merck (Darmstadt, Germany) (CAS 75-05-8). Water was purified using a Milli-Q system (Millipore, Billerica, MA, USA).

The ATP assay kit (119107) and LPO assay kit (437639) were from Sigma Aldrich (Sigma Aldrich, St. Louis, MO, USA).

### 4.2. Preparation of Genistein Solution

A genistein solution was prepared daily in water, diluting 500 mg of genistein in 10 mL of ultrapure water (50 mg/mL). This solution was kept at 4 °C since genistein is stable in water for 24 h [33].

### 4.3. Experimental Model

The study was conducted in 60-day-old Wistar rats (*Rattus norvegicus*) (male, adults), weighing 250–350 g. They were divided into 2 groups: control and treated group (*n* = 8 animals/group). The animals were kept in an environment with controlled temperature and humidity, availability of water and ad libitum diet, and light–dark cycle (12 h/12 h).

#### 4.3.1. Treatments

The animals received 1 mL of genistein solution or saline solution by gavage daily for 5 days. Genistein was diluted in saline solution (50 mg/mL) and administered by gavage. Genistein dose was settled following the daily intake considered clinically relevant [2,34,35].

The animals were placed in individual cages, allowing the individualized evaluation of body weight gain. In addition, it was observed if there were signs of toxicity (respiratory function, piloerection, diarrhea, cyanosis, and mucous pallor). These evaluations were made by the coordinating professor of the project, accredited as a competent person for animal experimentation and issued by the General Directorate of Veterinary Medicine of Portugal and by a veterinarian doctor. If there were any signs of intense clinical toxicity, the animals would be euthanized in advance. Otherwise, they were euthanized on the sixth day after administration of the experiment solutions [12,36].

To collect spermatozoa, anesthesia was performed in the animals with 5% isoflurane. After the completion of this step, euthanasia was performed with the administration of overdose of the same anesthetic [12,36].

This project was approved by the Ethics Committee on the Use of Animals of the University of Passo Fundo (protocol number 007/2022), and the procedures were performed according to the national guidelines for animal experiments (Brazilian Law 11.794/2008).

#### 4.3.2. Monitoring of Consumption

During the five days of treatment, the animals were evaluated for individual relative water and food intake and body mass gain.

#### 4.3.3. Euthanasia of Animals

Euthanasia of the animals took place after anesthesia for the collection of spermatozoa. For the procedure in the euthanasia chamber, 5% isoflurane (through a hospital gauze/cotton soaked in the anesthetic fluid, protected from animal contact) was associated with camera saturation. After euthanasia, the confirmation of death was carried out by a professional qualified for this purpose, confirming the loss of consciousness and vital signs [36].

#### 4.3.4. Relative Weight of Sexual Organs

After euthanasia, the sexual organs (right testicle, left testicle, right epididymis, left epididymis, seminal vesicle, ventral prostate, tail of the right epididymis, and tail of the left epididymis) were excised and weighed. The seminal vesicle was weighed, filled, and subsequently emptied. Results were expressed as the relative weight, considering the whole-body mass of each animal [12].

#### 4.3.5. Histological and Morphological Analysis

The left testicle was fixed in Bouin, embedded in paraffin, sectioned in a microton (5 μm), mounted on a slide, and stained with hematoxylin/eosin. Using the 20× objective, we searched for 100 seminiferous tubules in stages VII and VIII that presented rounded morphological aspects visualized in cross-section. Stage VII is characterized by elongated spermatids moving to the luminal surface of the seminiferous epithelium; the nuclei of round spermatids have not made contact with the cell surface and also the acrosome angle is greater than 120°. In stage VIII, the spermatid nucleus with its acrosome makes contact with the plasmatic membrane [27].

To obtain the thickness of the seminiferous epithelium and the tubular diameter, 100 rounded tubules in stages VII and VIII [12] were used, visualized by microscope (Zeiss Primo Star with AxioCam ERc 5s, Zeiss, Jena, Germany) under 200× magnification using ZEN (Zeiss—Blue Edition) software 3.7. Five seminiferous tubules from each group at stage VII, at a magnification of 400×, were used for counting viable and degenerated Sertoli cells, spermatogonia, primary spermatocytes, rounded, and elongated spermatids. Further, the Leydig cells were counted at adjacent intertubular compartments (space surrounding the analyzed tubule limited by adjacent tubules) [27]. Degenerative changes were considered to be (a) presence of elongated spermatids near the basal compartment, without sickle shape when viewed completely and with dense eosin staining; (b) nuclear retraction in round spermatids and discontinuity of their acrosome; (c) spermatocytes, spermatogonia, Sertoli, and Leydig cells with pyknotic nuclei [27].

#### 4.3.6. Testosterone Quantification

The determination of testosterone in plasma was performed by liquid chromatography–tandem mass spectrometry. Blood was collected through the puncture of the vena cava. After centrifugation, the plasma was separated and packaged for subsequent testosterone quantification. Samples were prepared in 1.5 mL plastic tubes with 100 μL of plasma, 10 μL of testosterone-*d*_3_ (10 μg/mL; internal standard), and 400 μL of acetonitrile to protein precipitation. After homogenization, samples were centrifuged at 9000× *g* for 6 min. Then, 400 μL of supernatant was collected following the addition of 1 mL of ethyl acetate and hexane mixture (60:40). After centrifugation at 9000× *g* for 6 min, the supernatant was collected and dried under a nitrogen stream at 40 °C, and the residue was reconstituted in 50 μL of acetonitrile, transferred to an auto-sampler vial, and an aliquot of 10 μL was injected into the analytical system.

A Nexera-i LC-2040C Plus system coupled to an LCMS-8045 triple quadrupole mass spectrometer (Shimadzu, Kyoto, Japan) was used for the analysis. The ESI-MS/MS parameters were set in positive ion mode as follows: 4500 V; desolvation line temperature, 250 °C; heating block temperature, 400 °C; drying gas, 10/L min; and nebulizing gas, 3 L/min. Collision-induced dissociation was obtained with 230 kPa argon pressure. Analyses were carried out with multiple reaction monitoring (MRM) by using the following fragmentations: *m*/*z* 289.20 → *m*/*z* 71.25; *m*/*z* 289.20 → *m*/*z* 56.85 e *m*/*z* 289.20 *m*/*z* 78.80 for testosterone ([M+H]^+^), and *m*/*z* 292.20 → *m*/*z* 109.10; *m*/*z* 289.20 → *m*/*z* 97.10 e *m*/*z* 289.20 *m*/*z* 79.20 for testosterone ([M+H]^+^). The chromatographic separation was conducted with a Shim-pack column (25 × 4.6 mm, 5 μm particle size) (Shimadzu, Kyoto, Japan) eluted with a flow rate of 0.3 mL/min. The gradient mobile phase system consisted of water (solvent A) and acetonitrile (solvent B), both fortified with 0.1% formic acid as follows: 0–4 min, 35–100% of B; 4–5 min, 100% of B; 5–5.10 min, 100–35% of B; 5–8 min, 35% of B. The column oven was kept at 30 °C. The data were processed using LabSolutions software 5.6 (Shimadzu, Kyoto, Japan).

#### 4.3.7. Seminal Collection

Rats’ spermatozoa were washed from the tail of the excised epididymis with a needle attached to a syringe containing isotonic salt medium constituted by 2 g/L BSA (Bovine Serum Albumin), 113 mmol/L KCl, 12.5 mmol/L KH_2_PO_4_, 2.5 mmol/L K_2_HPO_4_, 3 mmol/L MgCl_2_, 0.4 mmol/L of Ethylenediaminetetraacetic Acid (EDTA) and 20 mmol/L Tris adjusted to pH 7.4 with HCl. Samples were then processed for oxygraphic studies after hypotonic treatment [37].

### 4.4. Mitochondrial Assays

#### 4.4.1. Mitochondrial Respiratory Activity

The mitochondrial respiratory activity of spermatozoa was evaluated with a polarographic assay of oxygen consumption performed in sperm cells. Polarography measures the rate of change of oxygen concentration in a solution and, since oxygen is the last in the respiratory chain to accept electrons, provides a direct measure of mitochondrial activity [38].

Oxygen uptake by rat spermatozoa (2 × 10^7^ sperm cells) was measured by using a Clark-type oxygen probe (Hansatech oxygraph; Hansatech Pentney, King’s Lynn, UK). Oxygen consumption was measured in the presence of a solution of respiratory substrates (10 mmol/L pyruvate and 10 mmol/L malate) and 0.76 μmol/L of adenosine diphosphate (ADP) (107360, 240176 and A2754, Sigma Aldrich).

Sperm mitochondria respiration efficiency was measured using the RCR, which was calculated by dividing rate of oxygen uptake measured in the presence of substrates + ADP (V_3_) by rate of oxygen uptake measured with substrates alone (V_4_) [10].

#### 4.4.2. Mitochondrial Respiratory Complexes Activities

To evaluate the activity of mitochondrial respiratory complex activity, sperm cells were resuspended in 20 mmol/L potassium phosphate buffer (pH 7.0) to obtain a final concentration of 2 × 10^8^ spermatozoa/mL and then disrupted by freeze-thawing before analysis. 

The catalytic activities of mitochondrial complexes I, II, III, and IV were evaluated spectrophotometrically [37,39]. Complex I activity was determined following the decrease in absorbance at 340 nm, using decylubiquinone (D7911, Sigma Aldrich) as electron acceptor and NADH (N8129, Sigma Aldrich) as donor. Complex II activity assay was evaluated following the decrease in absorbance at 600 nm, using succinate (S3674, Sigma Aldrich) as donor and DCPIP (119814, Sigma Aldrich) as electron acceptor. Complex III and complex IV activities were determined by measuring the variation of absorbance at 550 nm, resulting from reduction and oxidation of cytochrome *c* (C2037, Sigma Aldrich), respectively. Catalytic activities expressed as nmoles/min.

#### 4.4.3. ATP and LPO Levels

ATP levels in sperm cells were determined by using the luciferase ATP assay kit and were expressed as nmoles of ATP/10^8^ spermatozoa [37].

Lipid peroxidation levels in sperm samples were determined using an LPO assay kit, which measures the redox reactions with ferrous ions. LPO was expressed as millimoles LPO/10^6^ spermatozoa [37].

### 4.5. Statistical Analysis

Data were expressed as mean ± standard deviation. Relative water and food intakes were evaluated using Two-Way ANOVA. Normality was assessed through Shapiro–Wilk test. Results were compared through Mann–Whitney or Student’s *t*-test, depending on data distribution pattern. Differences were considered statistically significant (*p* < 0.05).

## 5. Conclusions

Genistein treatment enhanced overall mitochondrial oxygen consumption, increased plasmatic testosterone levels, and presented a pro-spermatogenesis effect without systemic toxicity. Thus, the sperm reached the end of the process with a better mitochondrial profile.

We demonstrated that a short treatment with a relevant dose of genistein is effective to improve sperm mitochondrial efficiency. The study has provided some valuable insights and expressed optimism about its potential contributions to the field of human and animal research.

## Figures and Tables

**Figure 1 ijms-24-14260-f001:**
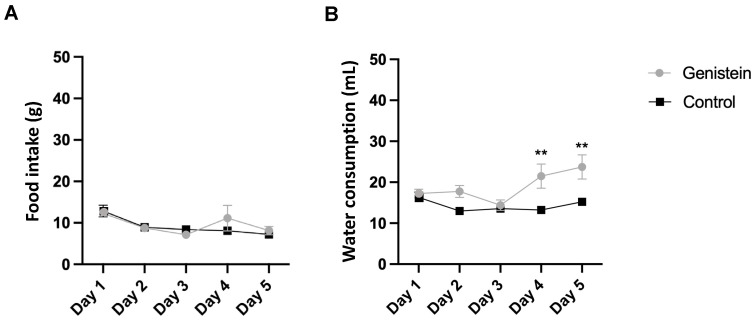
(**A**) Food and (**B**) water consumption. Results are presented as mean ± standard deviation (*n* = 8 animals per group). Statistical analysis was performed with Two-Way ANOVA (** *p* < 0.01 vs. control).

**Figure 2 ijms-24-14260-f002:**
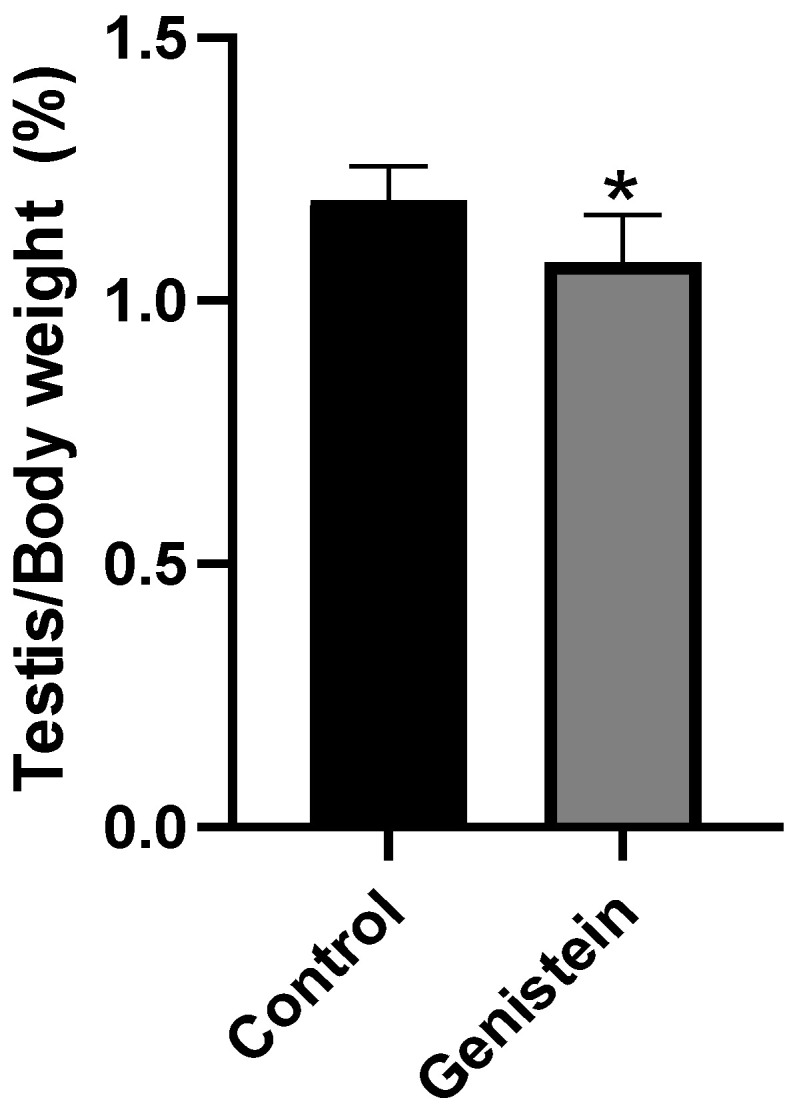
Relative mass of testicles. Results are presented as mean ± standard deviation from the sum of right and left testicles (Appendix A) normalized by the total body mass of each animal (*n* = 8 animals per group). Control group: 1.19 ± 0.06%. Genistein-treated group: 1.07 ± 0.08%. Statistical analysis was performed with Mann–Whitney test (* *p* < 0.05 vs. control).

**Figure 3 ijms-24-14260-f003:**
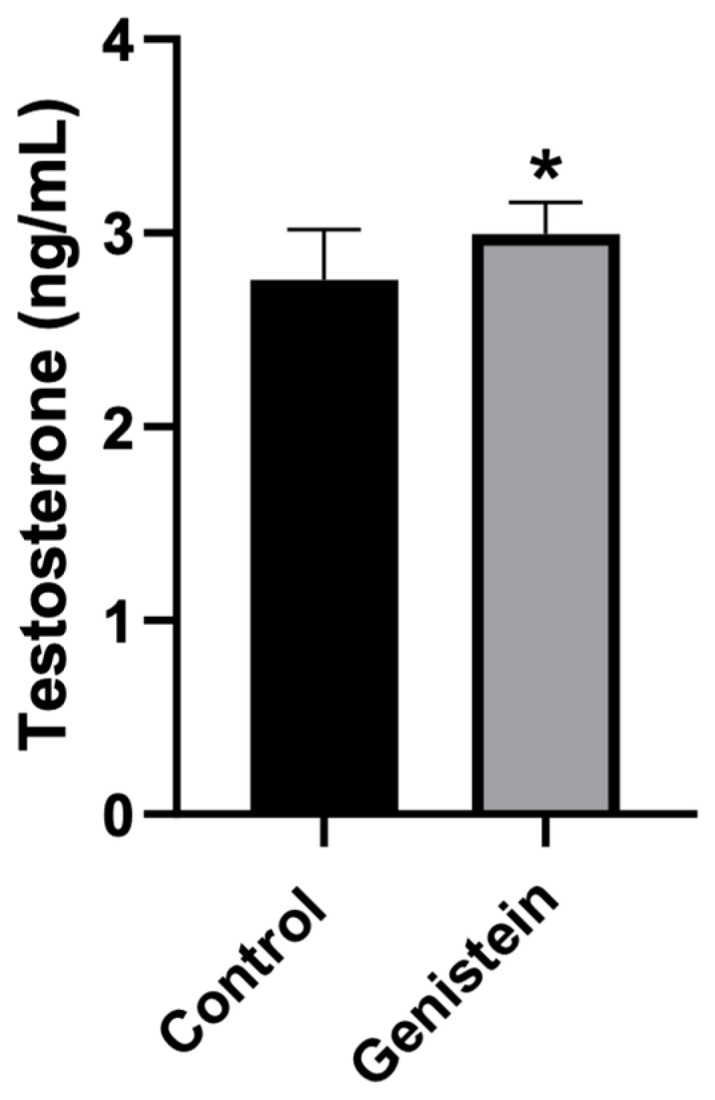
Plasma testosterone levels. Results are presented as mean ± standard deviation (*n* = 8 animals per group). Control group: 2.76 ± 0.26 ng/mL. Genistein-treated group: 3.00 ± 0.16 ng/mL. Statistical analysis was performed with Kolmogorov–Smirnov test and Student’s *t*-test (* *p* < 0.05 vs. control).

**Figure 4 ijms-24-14260-f004:**
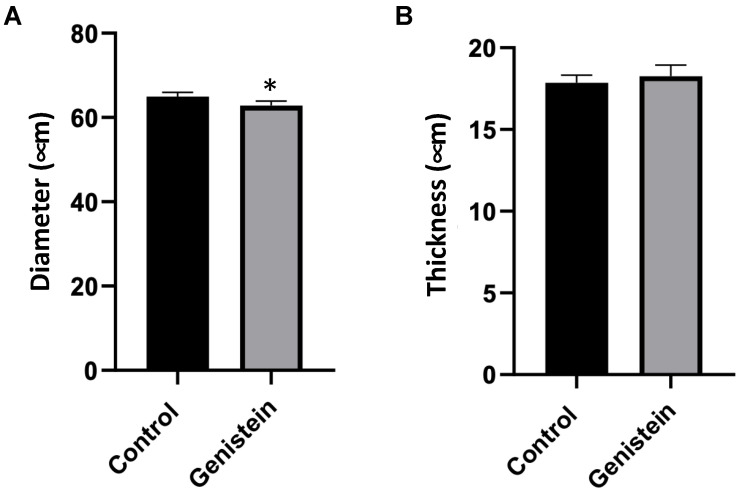
Seminiferous tubules cross-section. (**A**) Diameter and (**B**) thickness of seminiferous tubules. Results are presented as mean ± standard deviation (*n* = 5 animals per group). (**A**) Control group: 64.97 ± 6.12 µm. Genistein-treated group: 62.86 ± 5.76 µm. (**B**) Control group: 17.85 ± 2.38 µm. Genistein-treated group: 18.25 ± 2.23 µm. Statistical analysis was performed using the Shapiro–Wilk test (* *p* < 0.05 vs. control).

**Figure 5 ijms-24-14260-f005:**
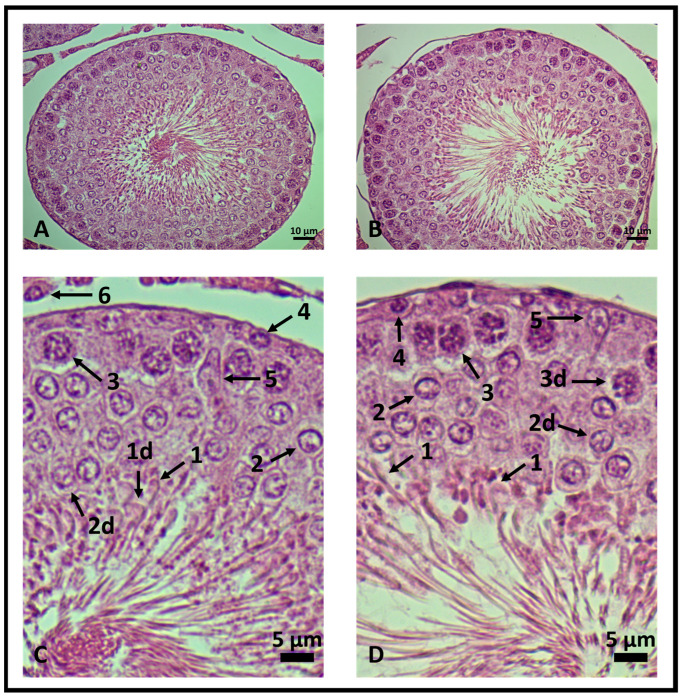
Testicular histology of rat seminiferous tubules. 400× magnification. (**A**,**C**) Control group and (**B**,**D**) genistein-treated group. In (**C**,**D**), the seminiferous epithelium is enlarged to highlight viable and degenerated cells (d). (1) Elongated spermatids, (2) rounded-shaped spermatids, (3) primary spermatocytes, (4) spermatogonia, (5) Leydig and (6) Sertoli cells (*n* = 5 animals per group).

**Figure 6 ijms-24-14260-f006:**
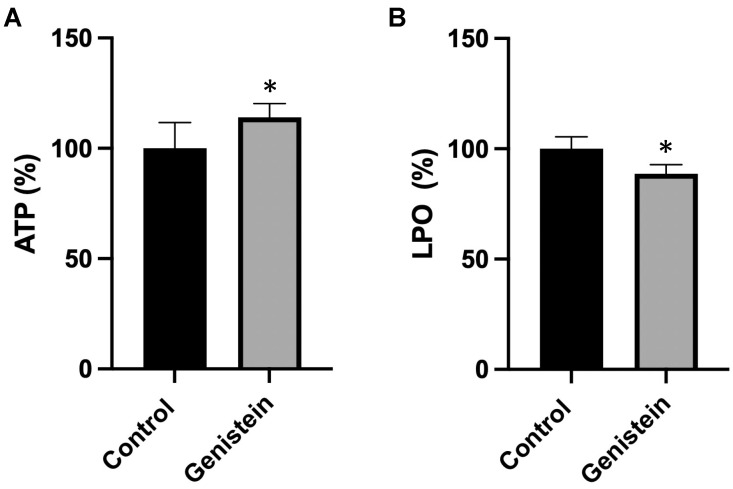
(**A**) ATP and (**B**) LPO levels. The amount of ATP or LPO in the control group was set to 100%. The values reported in the figure represent the mean ± standard deviation (*n* = 8 animals per group). (**A**) Control group: 100.0 ± 11.7%. Genistein-treated group: 114.4% ± 6.3%. (**B**) Control group: 100.0 ± 5.4%. Genistein-treated group: 88.7 ± 4.1%. Statistical analysis was performed using Student’s *t*-test (* *p* < 0.05 vs. control).

**Table 1 ijms-24-14260-t001:** Histopathological evaluation of testes.

Parameters	Cell Types	Groups	
		Control	Genistein
Viable cells	Elongated spermatid (c/t)	22.08 ± 8.82	9.8 ± 1.88 *
	Rounded-shaped spermatid (c/t)	22.48 ± 7.13	14.16 ± 2.15
	Primary spermatocytes (c/t)	37.84 ± 4.00	46.24 ± 4.26 *
	Spermatogonia (c/t)	22.76 ± 5.33	35.36 ± 3.39 **
	Sertoli Cells (c/t)	10.08 ± 1.78	13.64 ± 2.78 *
	Leydig Cells (c/aic)	35.12 ± 8.80	50.88 ± 10.19 *
Degenerated cells	Elongated spermatid (c/t)	48.72 ± 10.82	49.88 ± 4.92
	Rounded-shaped spermatid (c/t)	109.40 ± 7.90	119.28 ± 13.50
	Primary spermatocytes (c/t)	8.85 ± 1.30	15.32 ± 3.49 *
	Spermatogonia (c/t)	12.00 ± 2.94	18.24 ± 5.42
	Sertoli Cells (c/t)	4.52 ± 0.63	6.96 ± 1.04 **
	Leydig Cells (c/aic)	17.60 ± 6.71	21.26 ± 8.64

Cellular changes were scanned in 5 seminiferous tubules/rat in 5 animals each group (control and genistein-treated). C: cells; t: seminiferous tubule; aic: adjacent intertubular compartment. Results are presented as mean ± standard deviation. Statistical analysis was performed using the Shapiro–Wilk test (* *p* < 0.05 vs. control, ** *p* < 0.01 vs. control).

**Table 2 ijms-24-14260-t002:** Mitochondrial oxygen consumption assay.

	V_3_	V_4_	RCR
	(nmoles O_2_/min/mL)	(nmoles O_2_/min/mL)	
Control	7.49 ± 0.27	4.21 ± 0.39	1.79 ± 0.03
Genistein	7.93 ± 0.35 *	3.41 ± 0.37 *	2.35 ± 0.18 *

Data are presented as mean ± standard deviation (*n* = 8 animals per group). Student’s *t*-test was performed to detect significant differences between the control and the genistein group (* *p* < 0.05 vs. control).

**Table 3 ijms-24-14260-t003:** Respiratory chain complexes activity.

	Complex I	Complex II	Complex III	Complex IV
	(nmoles of NADH ox/min)	(nmoles of DCPIP red/min)	(nmoles of cyt *c* red/min)	(nmoles of cyt *c* ox/min)
Control	5.3 ± 0.3	8.7 ± 0.4	11.3 ± 0.7	6.3 ± 0.8
Genistein	5.4 ± 0.3	8.8 ± 0.4	12.5 ± 0.4 *	6.4 ± 0.5

Data are presented as mean ± standard deviation (*n* = 8 animals per group). Student’s *t*-test was performed to detect significant differences between the control and the genistein group (* *p* < 0.05 vs. control). NADH (Nicotinamide adenine dinucleotide reduced), DCPIP (2,6-Dichlorophenolindophenol), cytochrome *c* (cyt *c*).

## Data Availability

Not applicable.

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
