# Peer review of "Evaluation of Genistein as a Mitochondrial Modulator and Its Effects on Sperm Quality"

_ijms, 2023, doi:10.3390/ijms241814260_

Round 1
Reviewer 1 Report
The paper describes the study that investigated the effects of genistein, a phytoestrogen found in soy, on sperm viability in male Wistar rats after a short oral administration (50 mg/day for 5 days). The study focused on mitochondrial activity and its influence on sperm viability. The genistein-treated group showed a reduction in testicular mass, an increase in viable Leydig, Sertoli, spermatogonia and primary spermatocytes, and an increase in plasma testosterone levels. The study also notes an improvement in sperm mitochondrial efficiency and overall oxygen utilisation.
Minor issues:
Background information: The study insufficiently describes genistein, phytoestrogens and their known effects on the male reproductive system. The significance of the study would be contextualised by a more thorough literature review.
Discussion of the study's limitations: The study's weaknesses are not discussed in detail. It would improve the transparency and trustworthiness of the study if the limitations, such as the short treatment duration, were acknowledged.
Conclusions and implications: Discussion of the broader implications of the results for animal and human fertility and possible directions for future research could be included in the conclusions section.
Material and methods: The material and methods section is not comprehensive as the instruments and techniques used in the study are not described in sufficient detail. Readers would understand the study better if the experimental protocols were described in more detail.
Author Response
Point 1: Background information: The study insufficiently describes genistein, phytoestrogens and their known effects on the male reproductive system. The significance of the study would be contextualised by a more thorough literature review.
Response 1: We appreciate this remark. We improved introduction section to comply with the Referee suggestion (marked in yellow in the new version of the manuscript, lines 42-47; 52-53; 57-68; 79-83).
Point 2: Discussion of the study's limitations: The study's weaknesses are not discussed in detail. It would improve the transparency and trustworthiness of the study if the limitations, such as the short treatment duration, were acknowledged.
Response 2: Thank you for this comment. However, we respectfully disagree with the Reviewer view of the short treatment duration as a limitation of the study. In fact, we considered the short protocol as a great advantage for future clinical applications. A short protocol with relevant effects is desirable because it is related to decreased costs, increased treatment adherence and, consequently, success. We made this point clearer int the new version of the manuscript (205-208).
Point 3: Conclusions and implications: Discussion of the broader implications of the results for animal and human fertility and possible directions for future research could be included in the conclusions section.
Response 3: Thank you for this remark. We included new text in the conclusion section (lines 465-468).
Point 4: Material and methods: The material and methods section is not comprehensive as the instruments and techniques used in the study are not described in sufficient detail. Readers would understand the study better if the experimental protocols were described in more detail.
Response 4: We thank the Reviewer for his considerations and recognize the lack of detailed methodology, with item 4.3.5 being completely reformulated (lines 367-380).

Reviewer 2 Report
The development of spermatogenic cells is regulated by both intrinsic and extrinsic factors, of which the later include nutrients, hormones and growth factors that are provided by somatic cells in the immediate environment of germ cells or through long-range secondary effects via HPG axis. In this manuscript, the authors examined effects of genistein, an isoflavone that resemble estrogen-like structure, often contained in consumable plant seeds. They observed interesting phenotypes when administered genistein to animals (rats) for a short period of time (5 days), including the changes of spermatogenic cells, testosterone production and mitochondrial functions of spermatozoa, providing a potentially useful system to study the regulatory roles of the natural products on spermatogenesis and male fertility. However, the work described in the manuscript is still in its early stage of development, data presented are often in-complete and experiments need to be better designed, including, for example, proper controls, so that the results could be understood cohesively. The main conclusion that the authors draw from the current data is that genistein had improved mitochondrial functions in sperm, in terms of respiratory functions when applied to the animals in a fairly short period of time. However, sperm were not assessed for their changes of motility and fertility. The effects of genistein on spermatogenic cells were somewhat hard to understand when the data showed that cells prior to spermiogenesis stage were increased, whereas elongating spermatids and testicular weights were decreased. Mitochondrial effects were not examined for the testicular spermatogenic cells. Methods used were not sufficiently described, such as: how the diameter and thickness of seminiferous tubules were measured, how spermatogenic cells and how viable and dead cells at different stages were examined and counted. The presentation of the data should also be paid attention to, for example, no labeling for the Y-axis in Figure 2, values of mean +/- s.d should be given out for results presented in Figures 2, 3, 4 and 6. What the increased water consumption mean for genistein effects (Figure 1). Overall, more experiments should be considered on the dosage of genistein applied to the animals, whether and how mitochondrial functions change in spermatogenic cells at different stages, whether the genistein effects on mitochondrial are related to the development of spermatogenic cells and eventually, what the outcome of fertility is after animals were treated with genistein.
Some editing of the English language are required.
Author Response
Point 1: “However, the work described in the manuscript is still in its early stage of development, data presented are often in-complete and experiments need to be better designed, including, for example, proper controls, so that the results could be understood cohesively. The main conclusion that the authors draw from the current data is that genistein had improved mitochondrial functions in sperm, in terms of respiratory functions when applied to the animals in a fairly short period of time. However, sperm were not assessed for their changes of motility and fertility. The effects of genistein on spermatogenic cells were somewhat hard to understand when the data showed that cells prior to spermiogenesis stage were increased, whereas elongating spermatids and testicular weights were decreased.”
“Overall, more experiments should be considered on the dosage of genistein applied to the animals, whether and how mitochondrial functions change in spermatogenic cells at different stages, whether the genistein effects on mitochondrial are related to the development of spermatogenic cells and eventually, what the outcome of fertility is after animals were treated with genistein.”
Response 1: We agree with the Reviewer that this study does not exhaust the subject and opens new points to be clarified. However, in the present manuscript we present an in vivo short protocol aiming to improve sperm quality using genistein. We clarified this aspect in lines 205-208.
This treatment caused increased plasmatic testosterone levels and a pro-spermatogenesis effect without systemic toxicity in rats. Genistein administration stimulated the overall oxygen consumption, improving mitochondrial efficiency in rat seminal samples. Then, we think this study, per se, presents great scientific contribution on fertility issues.
Point 2: Methods used were not sufficiently described, such as: how the diameter and thickness of seminiferous tubules were measured, how spermatogenic cells and how viable and dead cells at different stages were examined and counted.
Response 2: We thank the Reviewer and recognize the lack of detailed methodology, with item 4.3.5 being completely reformulated (lines 367-380). Likewise, in Table 1 the term “dead cells” was replaced for “degenerated cells”. Additionally, we inform that the data presented in Table 1 represent absolute values obtained through cell counts of 5 seminiferous tubules in stage VII and respective adjacent intertubular compartment for counting Leidyg cells.
Point 3: The presentation of the data should also be paid attention to, for example, no labeling for the Y-axis in Figure 2, values of mean +/- s.d should be given out for results presented in Figures 2, 3, 4 and 6.
Response 3: All these minor revisions were fully attended.
Point 4: What the increased water consumption mean for genistein effects (Figure 1).
Response 4: We don’t place a high value on this result because it is likely attributable to the bitter taste of the genistein solution. We added this hypothesis in the new version of the manuscript (lines 287-289).

Reviewer 3 Report
the aim and the writing style is very good Please add more citation in the introduction section to be more detailed enough .in materials and methods what about the ethical statement in addition to the inclusion and exclusion criteria please mention???
the limitation of this study should be presented in a good form
the histological analysis must be rewritten again
All figures are small of very bad qulaity please correct it
Please delete some references from 15-20 and check reference number 4
the introduction is well written but more details must be provided
second paragraph in discussion must be reorganized
English must be improved
Author Response
Point 1: The aim and the writing style is very good.
Please add more citation in the introduction section to be more detailed enough.
The introduction is well written, but more details must be provided
Response 1: Thank you for this comment.
Introduction section was rewritten (changes marked in yellow).
Point 2: in materials and methods what about the ethical statement in addition to the inclusion and exclusion criteria please mention???
Response 2: Ethical statement is declared (lines 342-345; 476-479)
Point 3: the limitation of this study should be presented in a good form
Response 3: We added lines 465-468.
Point 4: the histological analysis must be rewritten again
Response 4: According to the Reviewer suggestion, this paragraph was rewritten (lines 367-380).
Point 5: All figures are small of very bad quality please correct it
Response 5: As suggested by the Reviewer, we improved figures quality.
Point 6: Please delete some references from 15-20 and check reference number 4
Response 6: The points raised by the Reviewer were fully attended in the new version of the manuscript (changes marked in yellow).
Point 7: second paragraph in discussion must be reorganized
Response 7: According to the Reviewer suggestion, we reorganized this paragraph (lines 199-208)

Reviewer 4 Report
The penomenon of phytoestrogen action on the male reproductive system is rather known, however some molecular mechanisms remains not well-studied. Here authors undertaken such studies however by routine analysis without deep look.
In the Abstract and throughout the text add „ cells” to Leydig and Sertoli
According to lines 53, 54 there is no detailed information on that, studies on animals and humans with the use of various systems were not presented
Lines 62-65, here information should be also more detailed
How did you estimate the number of dead cells?
The histology shows only one tubule. Please present full testicular tissue. Also, cross-sections of the epididymis should be presented. Smears of spermatozoa are not shown either.
It will be nice to have transmission electron microscope analysis of mitochondria or immunohistochemical/western blot data.
In the Discussion, you should give many more examples of how other phytoestrogens, not only genistein, affect on studied by you parameters
Author Response
Point 1: In the Abstract and throughout the text add „ cells” to Leydig and Sertoli
Response 1: Requested change is added in the new version of the manuscript (lines 30, 149, 192).
Point 2: According to lines 53, 54 there is no detailed information on that, studies on animals and humans with the use of various systems were not presented.
Lines 62-65, here information should be also more detailed.
Response 2: We appreciate this remark. We included detailed information requested in the new version of the manuscript (lines 57-68).
Point 3: How did you estimate the number of dead cells?
The histology shows only one tubule. Please present full testicular tissue. Also, cross-sections of the epididymis should be presented. Smears of spermatozoa are not shown either.
Response 3: We thank the Reviewer for his considerations, and we recognize the lack of detailed methodology, with item 4.3.5 being completely reformulated (lines 367-380). Likewise, in Table 1 the term “dead cells” was replaced for “degenerated cells”. Additionally, we inform that the data presented in Table 1 represent absolute values obtained through cell counts of 5 seminiferous tubules in stage VII and respective adjacent intertubular compartment for counting Leidyg cells.
Point 4: It will be nice to have transmission electron microscope analysis of mitochondria or immunohistochemical/western blot data.
Response 4: We appreciate the Reviewer suggestion. It is our intention to consider this suggestion in our future studies.
Point 5: In the Discussion, you should give many more examples of how other phytoestrogens, not only genistein, affect on studied by you parameters
Response 5: Thanks for this comment. We added more examples of other phytoestrogens affect the parameters evaluated in the present manuscript. (211-212; 263-264; 285-304).

Round 2
Reviewer 2 Report
The authors made improvements in this version. However, for the current data, some key information is still missing; the presentation of the data, including the figures’ resolution should also be improved, specifically:
1) The method the authors used to count cells should be described clearly in the Materials and Methods: did the authors use morphological differences to tell which cells are at what stage (including viable or dead cells) for the results of Table 1? The methods should be described clearly in Section 4.3.5.
2) Figure 2: Label the Y-axis as “Testis/Body weight %”. What are the weights of testes in each group, show a graph for the comparison of testis weights, you should have the data.
3) In the legends for Figure 2, 3, 4, 6, use mathematic formula to show the results, instead of word, e.g. Control group: 1.19+/-0.06, Genistein group: 1.07+/- 0.08, with proper units.
4) Table 1: give out the unit of the numbers, is it number of cells/per seminiferous tubule section? The number of Sertoli and Leydig seemed increased in Genistein group, but so as the degenerated Sertoli and Leydig cells, discuss this in the Discussion.
5) In Figure 5, uses arrows or arrowheads (different colors maybe) to indicate the different cell types, so that the readers could see which cells are counted as Sertoli, Leydig, etc.
6) Materials and Methods: give out catalog numbers for the items purchased from different companies,
7) Materials and Methods: Lines 378-380, mentioned comparison of sperm morphologies, what are the results? Show these results in a figure. Presumably this is different from the seminal samples collected for mitochondrial assays (lines 411-416), did the seminal samples collected from the epididymis contain other cell types than sperm? Indicate clearly in the Results (section 2.6) that it was the seminal sperm from epididymis used for metabolic measurements.
Some editing needed.
Author Response
Point 1: The method the authors used to count cells should be described clearly in the Materials and Methods: did the authors use morphological differences to tell which cells are at what stage (including viable or dead cells) for the results of Table 1? The methods should be described clearly in Section 4.3.5.
Response 1: As requested, the cell counting method was detailed in item 4.3.5, including the characterization of stages VII and VIII (lines 377-381), as well as degenerative cellular changes (lines 388-393).
Point 2: Figure 2: Label the Y-axis as “Testis/Body weight %”. What are the weights of testes in each group, show a graph for the comparison of testis weights, you should have the data.
Response 2: We changed the label of the Y-axis as solicited. Considering the weights of testis, we prefer to express these results normalizing the weight of testis with the total body mass of the rat. It is more correct because a larger animal will have larger organs and this variation is overcome with normalization. Even so, aiming to attend the referee’s point, we suggest making the requested graph available as a supplementary material (line 108, last page of the manuscript).
Point 3: In the legends for Figure 2, 3, 4, 6, use mathematic formula to show the results, instead of word, e.g. Control group: 1.19+/-0.06, Genistein group: 1.07+/- 0.08, with proper units.
Response 3: This point was fully attended (lines 109, 116, 130-131, 186-187).
Point 4: Table 1: give out the unit of the numbers, is it number of cells/per seminiferous tubule section? The number of Sertoli and Leydig seemed increased in Genistein group, but so as the degenerated Sertoli and Leydig cells, discuss this in the Discussion.
Response 4: In Table 1, the units referring to the counts of each cell type were included and indicated in the legend as follows: c: cells; t: seminiferous tubule; aic: adjacent intertubular compartment (line 141).
We have also included a possible hypothesis regarding the increase in Sertoli/Leydig cells in the discussion section (line 246-252).
Point 5: In Figure 5, uses arrows or arrowheads (different colors maybe) to indicate the different cell types, so that the readers could see which cells are counted as Sertoli, Leydig, etc.
Response 5: As requested, Figure 5 was improved. Arrows were included indicating the different cell types counted, which were referenced in the new legend of Figure 5 (lines 147-150).
Point 6: Materials and Methods: give out catalog numbers for the items purchased from different companies,
Response 6: This point was fully attended (lines 314, 317, 319, 320, 442, 454, 455, 456, 457, 459).
Point 7: Materials and Methods: Lines 378-380, mentioned comparison of sperm morphologies, what are the results? Show these results in a figure.
Presumably this is different from the seminal samples collected for mitochondrial assays (lines 411-416), did the seminal samples collected from the epididymis contain other cell types than sperm? Indicate clearly in the Results (section 2.6) that it was the seminal sperm from epididymis used for metabolic measurements.
Response 7: The comparison of sperm morphology, referring to the washing of the vas deferens, was removed from the manuscript due to flaws in the processing of the slides, which made an adequate analysis unfeasible.
In accordance with the Reviewer's comment, we have clarified that the semen samples collected for the mitochondrial studies were epididymal spermatozoa (lines 152-154).
Although the semen samples collected from epididymis may contain cell types other than spermatozoa, in our experimental system only sperm mitochondria do not swell in the hypotonic solution (Keyhani & Storey, 1973; Piasecka et al., 2001; Ferramosca et al., 2008), because the sperm outer mitochondrial membrane has different physical and chemical parameters than the somatic mitochondrial membrane.

Reviewer 4 Report
I accept corrections
Author Response
We thank the Reviewer 4 for his comments and suggestions which allowed us to significantly improve the manuscript.